# On the Comparison of Records from Standard and Engineered Fiber Optic Cables at Etna Volcano (Italy)

**DOI:** 10.3390/s23073735

**Published:** 2023-04-04

**Authors:** Sergio Diaz-Meza, Philippe Jousset, Gilda Currenti, Christopher Wollin, Charlotte Krawczyk, Andy Clarke, Athena Chalari

**Affiliations:** 1Department of Geophysics, GeoForschungsZentrum-Potsdam (GFZ) Telegrafenberg, 14473 Potsdam, Germany; 2Institute of Applied Geosciences, Technical University of Berlin, Ernst-Reuter-Platz 1, 10587 Berlin, Germany; 3Instituto Nazionale di Geofisica e Vulcanologia (INGV), Piazza Roma 2, 95125 Catania, Italy; 4Silixa Limited, 230 Centennial Park, Elstree WD6 3SN, UK

**Keywords:** distributed dynamic strain sensing, distributed acoustic sensing, stacking, saturation, multi-fiber optics

## Abstract

Distributed Dynamic Strain Sensing (DDSS), also known as Distributed Acoustic Sensing (DAS), is becoming a popular tool in array seismology. A new generation of engineered fibers is being developed to improve sensitivity and reduce the noise floor in comparison to standard fibers, which are conventionally used in telecommunication networks. Nevertheless, standard fibers already have extensive coverage around the Earth’s surface, so it motivates the use of the existing infrastructure in DDSS surveys to avoid costs and logistics. In this study, we compare DDSS data from stack instances of standard multi-fiber cable with DDSS data from a co-located single-fiber engineered cable. Both cables were buried in an area located 2.5 km NE from the craters of Mt. Etna. We analyze how stacking can improve signal quality. Our findings indicate that the stack of DDSS records from five standard fiber instances, each 1.5 km long, can reduce optical noise of up to 20%. We also present an algorithm to correct artifacts in the time series that stem from dynamic range saturation. Although stacking is able to reduce optical noise, it is not sufficient for restoring the strain-rate amplitude from saturated signals in standard fiber DDSS. Nevertheless, the algorithm can restore the strain-rate amplitude from saturated DDSS signals of the engineered fiber, allowing us to exceed the dynamic range of the record. We present measurement strategies to increase the dynamic range and avoid saturation.

## 1. Introduction

Optical fibers are able to carry light and propagate it over long distances with minimum attenuation, and so is a preferred medium for broad-band transmission systems around the world. When using the scattering properties of the glass fibers, the fiber can be turned into a distributed sensing system that provides substantial advantages over conventional electronic sensors [1,2,3,4]. Optical fiber sensors are able to measure external environment variables (e.g., temperature and strain; [5]) along their length with fine spatial intervals. The different natural light scattering within the fiber, Rayleigh, Brillouin, and Raman amongst them, are sensitive to strain and temperature [6,7,8,9].

Distributed Dynamic Strain Sensing (DDSS), also known as Distributed Acoustic Sensing (DAS), takes place when the Rayleigh scattering is used aside with optical time-domain reflectometry (OTDR; [7,10]) in a distributed optical sensor. This allows the monitoring of the strain rate along an optical fiber at high frequencies, with long distances, and with a high spatial interval [6,11]. Research in DDSS extends to applications such as borehole, infrastructure and urban activity monitoring [12,13,14], structure imaging [6], and natural hazards [15,16,17], amongst others.

Measurements in DDSS are carried out conventionally by using standard fibers. Light pulses are sent along the fiber, and the measurements depend on the phase of the back-scattered signals produced by random imperfections within the glass fibers. Phases in consecutive time intervals are compared in the extremes of a defined length, named gauge length, to receive a phase-shift measurement. The phase shift is an expression of the strain rate within the defined gauge length.

Proper phase readings depend on the intensity of the back-scattered signal, and so does the signal-to-noise ratio (SNR). However, some of the imperfections in the glass do not guarantee a uniform high-intensity scattering of the light, so it affects the dynamic range and introduces noise [18,19]. In consequence, engineered fibers are made with the purpose of overcoming this challenge in DDSS deployments. While standard fibers are best designed to transmit information, engineered fibers are made to intensify back-scattered light while probing as much length as possible. It consists of setting precise scattered centers (imperfections) uniformly distributed along the fiber [18], which allows them to achieve up to 100× (20 dB) improved sensitivity compared to conventional standard fibers [11,20,21]. Despite this, both fibers can exhibit distortion in their signal due to strong motion events or high-frequency events, as there is a maximum measurable strain limited by survey configurations [22]. The distortions are interpreted as signal saturation, and optical noise can reduce the chances of interpreting them.

There is a noticeable improvement in the DDSS methods with the use of engineered fibers in terms of SNR and sensitivity [11,18,20,23]. Nevertheless, the increase in back-scatter light intensity compromises the maximum possible length to interrogate. In addition, conventional standard fibers already have extensive coverage around the Earth’s surface and cities, so there is an interest to make use of the existing telecommunication infrastructure for measurements [1,14,24,25,26].

Conventional telecom cables often have several fibers (12 to 80), but distributed fiber optic sensing methods only use one dedicated fiber. Using two or more fiber instances of the same cable may allow an increase in the SNR, e.g., by stacking each fiber’s instance measurements [27]. While the expected signal is the same, random optical noise could be reduced. In this study, we aim to exploit the use of a multi-fiber cable and explore related processing methods by using several standard fibers within a standard fiber optic cable instead of just one, as the conventional way, and compare its performance with respect to an engineered fiber optic cable. To address this, we deployed a cable comprising 12 individual standard telecom fibers and a cable comprising an engineered fiber (commercial name “Constellation” from Silixa) at Mt. Etna volcano, Sicily. Etna is an ideal place to perform such an approach as it is well known for its frequent volcanic activity. In the area, natural signals cover a wide frequency range (0.05–100 Hz), giving an opportunity for testing DDSS in complex seismo-acoustic environments [6,15,16,28].

We first explain our setup in the field, then explore our proposed technique to spatially match the different standard fibers, and expand this method to match the standard fibers with the engineered fiber. Following, we introduce and characterize saturations found within the DDSS data and a proposed method to assess them. In the end, this leads to a performance analysis between the use of multiple standard fibers and single engineered fibers.

## 2. Materials and Setup

We deployed two fiber optic cables (non-helical) at a 2.5 km distance from the active craters of Mt. Etna volcano in Sicily and buried them beneath the ground in non-consolidated scoria. The first cable is 1.5 km long, contains 12 individual telecom fibers (standard), and was buried 30 cm beneath the surface (July 2019). The second cable is 0.5 km long, contains an engineered fiber (Constellation), and was buried at 15 cm depth in another trench following the same path as the standard cable (September 2019). Figure 1 shows the spatial configuration of both cables, where the engineered cable (blue) only covers one section (orange line) of the total length of the standard cable (black).

For simplicity of the fieldwork, we connected the engineered fiber to one of the 12 fibers within the standard cable (see Figure 2) and recorded only the section with the engineered fiber, ignoring the standard fiber section. In order to exploit the use of standard multi-fibers and compare it with the engineered ones, we spliced nine individual fibers of the standard cable between them in a back-and-forward style, as shown in Figure 2, and left two unused. For future reference, each of the fibers involved in the standard back-and-forward connections is called a “fiber instance” from 1 to 9 (see Figure 2). As a result, the standard fiber length is extended up to 13.5 km, probing the same geographic trajectory nine times (cable geometry; Figure 1).

The used engineered fiber is composed of the same glass material as the standard one but with scattering centers. Each scattering center works as a mirror that reflects around 0.01% of the transmitted light, increasing the back-scattered light intensity by 100 times more the one of the standard fibers [29,30]. The scattering centers are spaced every 5 m along the fiber, in this case, to match half of the used gauge length (10 m) [18,29,30].

Each type of fiber was recorded with a different interrogator [31] (Silixa instruments, Elstree, UK). The standard fiber (13 km) is sampled with an iDAS interrogator (intelligent Distributed Acoustic Sensor) [32]. The engineered fiber (0.5 km) is sampled with a dedicated interrogator called Carina, a unit designed to exploit the enhanced fiber performances [11]. We connected the iDAS to the first standard fiber instance and the Carina to an individual standard fiber of the same cable but spliced at the end of the standard cable to the engineered fiber (Figure 2). However, as previously stated, only the section where the fiber is engineered was recorded. Both interrogators were sampling at 1 kHz, with a gauge length of 10 m, but with different virtual sensor (channel) spacing: standard fiber with 2 m and engineered with 1 m between channels.

## 3. Methods

### 3.1. Location of the Channels

In order to locate each channel (trace) within the known path of the cable, we performed tap tests at specific points along the cable [6,15,24] with precise location (GPS measurements; see Appendix A) and associated the channel trace with the first arrival of a wave to the location of the tap (Figure 3). The location of the intermediate channels was computed through linear interpolation following the cable path.

### 3.2. Standard Fiber Stacking

To compare the signals recorded in the standard fibers, we need to make sure that the channels from the different fiber instances match spatially as best as possible (Figure 2). As shown in Figure 3, we divide the raw data into windows. Each window is a subset of the 2D spatio-temporal recording matrix representing a fiber instance as the ones shown in Figure 2.

To find the best channels correspondence, we cross-correlate (CC) with each other the windows corresponding to each individual fiber instance along the spatial dimension (Figure 3). Since the back-and-forward interconnections change the direction of how the fiber was probed, even windows are flipped. Let f(x,t) represent the 2D spatio-temporal recording matrix, where $f_{1}\left(x,t\right),f_{2}\left(x,t\right),\ldots ,f_{N}\left(x,t\right)$ describe the 1,2,…,N subsets previously defined as windows. Then, for any specific time sample $t_{0}$, a spatial function $f_{1}\left(x,t_{0}\right)$ describes the strain rate along the distance *x* of one fiber instance. For the same $t_{0}$, a function for a second fiber instance in the cable $f_{2}\left(x,t_{0}\right)$ is defined. The spatial functions of both fibers vary with *x*, but where the channels of the fibers share the same location, they should be the same spatial function. Therefore, the two spatial signals are cross-correlated to produce a CC function $\left(f_{1}\left(x,t_{0}\right)*f_{2}\left(x,t_{0}\right)\right)$.

The spatial CC is repeated for all time samples, so then all CC functions can be stacked and produce a final spatial cross-correlation function $S_{CC}$, as described by the following equation:(1)$$S_{CC}\left(1,2\right)=\sum_{i=0}^{T}CC_{i}=\sum_{i=0}^{T}\left(f_{1}\left(x,t_{i}\right)*f_{2}\left(x,t_{i}\right)\right)$$
where *T* is the total number of time samples in a given time window. The lag where the stacked CC function reaches its maximum value is used to shift the positions of the channels and establish a channel-to-channel correspondence between the two fiber instances.

We use the first fiber as a base against the other instances, compute the spatial CC to find the best channel equivalences, and stack the signals (dependent on time) by summing per channel and averaging them. The stacks have instances in the same way as the fiber instances. We define a stack instance *N* as the consecutive stack of fiber instances; from 1 to *N*.

### 3.3. Standard and Engineered Channel Equivalences

When comparing the standard and engineered fibers, the channel correspondence must be addressed similarly. We applied the same routine of spatial CC to have channel equivalences between both types of fibers. For this purpose, we used the standard fiber instance one and the engineered one (Figure 2). The common section restricts some length of the engineered fiber, and thus only the channels in the range of 183 to 493 was used, as shown in Figure 1. The DDSS data from the engineered fiber have to be flipped since this fiber was probed in the opposite direction with respect to the standard one.

As mentioned before, both types of fibers were interrogated with different channel spacing; 2 m for standard and 1 m for engineered. To compare signals from both cables, we explore two methods: (1) by introducing artificial channels with linear interpolation between the existing ones in the standard DDSS data and (2) by removing intermediate channels in the engineered DDSS data. The two methods differ by a shift of one channel, which then assures that the spatial CC works properly. We decided to adopt the alignment obtained by the second method since this one depends exclusively on real DDSS data.

To analyze the relation between standard fiber instances, stack instances, and engineered fibers, we performed a Principal Component Analysis (PCA) on the amplitudes of each of the stack instances at the standard fiber in relation to the amplitudes of the engineered fiber. This method constructs a covariance matrix using a set of variables. The retrieved eigenvalues and eigenvectors are used to analyze multi-variable relationships for a subsequent variable reduction to re-describe the relationship in lower dimensions [33,34]. In our case, we used the standard and engineered channel equivalence strain rate amplitudes for a given pair of equivalent channels as variables. The eigenvalues and eigenvectors describe the magnitude and directions of the maximum and minimum variance, respectively. The largest eigenvalue $\lambda_{1}$ and associated eigenvector $\vec{e_{1}}$ relate to the linear relation between the standard and the engineered signal amplitudes. The smallest eigenvalue $\lambda_{2}$ and associated eigenvector $\vec{e_{2}}$ would be related to divergences orthogonal to $\vec{e_{1}}$ and correspond to the noise [34].

### 3.4. Signal Saturation Correction

DDSS signals could be saturated due to strong motion events, such as tap tests. Signal saturation in DDSS is not as similar as in Broad-Band (BB) seismic instruments (Trillium Compact Sensors, Nanometrics Ltd., Kanata, ON, Canada). In BB instruments, the saturation is shown as $\pm V_{max}$ values where the signal exceeds the dynamic range maximum value $\pm V_{max}$. DDSS systems measure back-scattered light phases in response to longitudinal fiber length changes. In the case of saturation in DDSS, values of the signal exceeding a certain dynamic range maximum value ${\dot{\epsilon }}_{max}$, either on upper or lower limits, will suffer phase jumps. Some of these jumps are from $-{\dot{\epsilon }}_{max}$ to ${\dot{\epsilon }}_{max}$ or vice-versa, depending on the trend of the signal before the saturation point.

Figure 4 shows examples of signal saturation due to strong motion for standard fiber (a,c) and engineered fiber (b,d). We recognize two types of features: (1) complete phase-jump situations, indicated by red arrows, and (2) quasi-phase jumps, indicated by purple arrows. While phase jumps cross all of the dynamic range, and so they indicate saturation, the quasi-phase jumps fail to achieve a complete phase jump, and so they appear as high-frequency noise in the signal maximum and minimum parts.

We attempt to correct the saturated signals by first computing the Hilbert Transform H[u(t)] since it contains the magnitude and phase information of the original signal. We calculate the angle from the imaginary part *ℑ* of *H* and take the derivative signal to have the angle change in time:(2)$$\begin{array}{c} d\theta =\frac{d}{dt}\left(Angle\left(ℑ\left(H\right)\right)\right) \end{array}$$

To restrict the values of dθ between −π and π, we add or subtract 2π to any value out of this range. Since this detects abrupt phase changes of any type, it will also enhance the quasi-phase jumps.

In this work, we focus on correcting for saturation due to phase jump since they can distort the real amplitude signal, and so any pre-processing, such as filtering, would not give the correct waveform (example in Appendix A). Therefore, we normalize dθ, divide by the real part of *H* and the maximum θ, and sum the derivative of the raw signal. The output gives peaks that mark phase jumps with their polarities. We name this output the phase-jump characteristic function (CF), and by using a defined threshold value in positive and negative, it will mark the time when a phase jump occurs. All values that are greater or lesser than the positive and negative threshold values will be set to zero, respectively.

We apply the phase-jump correction to both standard and engineered fiber DDSS data on two different local seismic events for each type of fiber since no strong seismic event was registered during the common operation dates of both fibers. As shown in Figure 4, phase jumps look similar to the high-frequency signal of the tap test, so the method could not discriminate between the tap test signal and phase jumps. Therefore, tap test signals were not considered for corrections.

## 4. Results

### 4.1. Fiber Signals Relation

The spatial CC method gives the best channel-to-channel correspondence between the standard fiber instances, so then we have a fair comparison for time series between each of the fibers. Figure 5a shows a tremor signal recorded by equivalent channels between the standard fiber instances. The amplitude range of the signal as well as the noise increases toward higher fiber instances.

In relation, Figure 5b shows the standard deviation (Std. Dev.) computed for the same time window of the tremor signal on each individual standard fiber instance. The standard deviation between the fiber instances varies, and it shows a tendency to increase toward higher fiber instances (proportional to fiber distance). However, in general, after fiber instance six, the standard deviation grows exponentially.

We also use spatial CC to find the best channel equivalences between standard and engineered fibers. Figure 5a also shows the comparison of the raw signal recorded between standard (black line) and engineered (yellow dashed line) fibers, with channels 336 and 261 as the equivalents between standard and engineered fibers, respectively. The signals have a similar phase, however, the engineered fiber registers a higher amplitude with pronounced peaks and lower noise than the standard fiber.

In addition, in Figure 6, the spectra of the same tremor signal for standard and engineered fibers indicate that, for low frequencies (<125 Hz), the engineered fiber displays almost double the amplitude of the standard fiber. However, for higher frequencies (>125 Hz), the spectral amplitude of the engineered fiber diminishes toward higher frequencies, while the standard remains flat.

### 4.2. PCA for Signal-Noise Analysis

To reduce the high-noise levels on raw data from standard fibers shown in Figure 5, we used the channel equivalences to stack the standard fiber instances. We analyze the tremor signal using a PCA (for more description, see Section 3.3). The main slope and the noise level are given between stacked instances of the standard fibers and the engineered fiber.

As an example, Figure 7 shows cross-plots between the amplitudes of each of the stack instances against the amplitudes of the engineered fiber. The PCA method gives eigenvalues and eigenvectors that describe the relationship between variables that compose the cross-plots. Eigenvectors $\vec{e_{1}}$ and $\vec{e_{2}}$ describe the major and minor axes directions of an ellipse-shape composed from the data points in the cross-plots (blue dots), respectively. The largest eigenvalue $\lambda_{1}$ and associated eigenvector $\vec{e_{1}}$ (yellow) is related to the linear relation between the two fiber signals. The slope can be obtained from the direction of $\vec{e_{1}}$. The smallest eigenvalue $\lambda_{2}$ and associated eigenvector $\vec{e_{2}}$ are related to the signal noise level. $\lambda_{2}$ is an indication of the noise level, as its associated eigenvector $\vec{e_{2}}$ points orthogonally to the linear relation.

Based on Figure 6, we established a value of 200 Hz to discriminate between the main frequency content of the signal (<125 Hz), and that of the noise (>125 Hz). We run the PCA analysis for a lowpass and highpass filtered signal. The noise level variations for standard fiber signals stabilize at 200 Hz, so this value was picked as the filter limit. We then expected some of the optical noise to influence the lowpass filter PCA analysis, but not substantially.

Figure 8 shows the variations of $\lambda_{2}$ and the slope for different fiber relations in lowpass and highpass filtered signals: standard stack instances vs. engineered fiber (a,b), standard fiber instance one vs. other standard instances (c,d), and standard fiber instance one vs. standard stack instances (e,f). Other cross-plots can be found in the Appendix A.

For the main frequency of the signal (lowpass), the comparison between the standard stack instances and the engineered fiber amplitudes shows a starting ratio of 1.641 and a $\lambda_{2}$ value of $7.24\times 10^{-15}$ for stack instance one (fiber instance 1) and at stack instance five, it reaches a minimum value of $6.03\times 10^{-15}$ and a ratio of 1.685 between the two signals. Between fiber instance one and the other instances, the variation of the slope is near 10% and not stable. Nevertheless, the $\lambda_{2}$ value is always increasing. The value seems stable up to stack instance five and then starts to increase again. Finally, for fiber instance one against stack instances, the $\lambda_{2}$ keeps increasing while the slope decreases up to stack instance four and then increases again.

For the noise frequency (highpass), the slopes between stack instances and engineered signal are three orders of magnitudes lower than the ones registered with the lowpass filtered signal. Further, the slope is indirectly proportional to $\lambda_{2}$. This proportionality applied for the other comparisons in the highpass filter; fiber instance one vs. other fiber instances, and fiber instance one vs. stack instances, and we also notice a decrease in the slope near 0 for higher fiber and stack instances in comparison to instance one.

Based on the results of Figure 8a, in Figure 9, we illustrate the comparison between fiber instance one DDSS data and the fifth stack instance of the standard fibers for the same tap test signal example as in Figure 4. The stacked DDSS data (b) show that random high and low strain values within the wavefield are removed when comparing against the first fiber instance (a) at the different interest areas of the DDSS data (green boxes 3, 4, and 5). Further, the stacked version smooths the data, and enables to observe an easy continuation of the wave phases (green boxes 1 and 2).

### 4.3. Signal Saturation Correction

We apply the suggested algorithm for signal saturation correction to three different types of traces: (1) stack instance four, (2) fiber instance one (stack 1), and (3) engineered fiber. Figure 10a–c shows the original signals in black with the corrected signals in green. Figure 10d,e shows the CF for stack and engineered DDSS data in purple.

We expected that a noise reduction due to stacking could facilitate the signal saturation corrections on standard fiber data. However, the correction performed for stack five and fiber instance one (Figure 10a,b) have problems since the main trend does not maintain and clearly affects the signal values. On the other hand, the algorithm applied to engineered fiber (Figure 10c) maintains the same trend as the original signal and shows clear phase-jump corrections (Figure 10e).

Regarding the CF, there is a clear difference between the one produced by the stacked DDSS data (Figure 10d) and the engineered (Figure 10e). The CF of stack instance five has more variations with less defined main peaks on the expected phase jumps, while the CF of the engineered one has better-defined peaks in the phase jumps in comparison to the rest of the CF values.

## 5. Discussion

### 5.1. Strain-Rate Difference in Standard and Engineered Fibers

Theoretically, two fibers deployed at the same location following the same path should measure the same strain rate. Nevertheless, we noticed a discrepancy of 66% in amplitudes for equivalent channels between the standard and engineered fibers (Figure 5a). The glass refraction index cannot be a factor since it is the same for both fibers, according to the metadata (see Appendix A). We discard any coupling problems with the scoria granular media [15], taking into account the difference in burial depth and distance. For the frequency range expected from signals in seismic environments (0.1–100 Hz; [35]) an almost perfect mechanical coupling can be assumed between the cable and the medium [36]. Instead, we attribute the amplitude difference to the structural differences between the cables. The rigidity, jacket, and gel surrounding the fiber are different between the two fiber cables, so the sensitivity could be different [37]. Due to the shear effects in the protecting layers around the fiber, the strain could not be the same as the host material where the cable is deployed. Therefore, the strain-rate reading must take into account the influence of the cable structure [36,38].

### 5.2. Standard Multi-Fiber Stack and Improvement

Optical noise is random across the length of the interconnected fiber instances, and so we expected to have a reduction during the stacking process, where the engineered fiber acts as a reference to evaluate the noise reduction during stacking. Effectively, from stack instance one to two there is a significant reduction in optical noise compared to other stack instances based on $\lambda_{2}$ values (see Figure 8a), also supported by the theory (see Appendix B and Appendix A).

An increase in noise can be noticed in stack instance four. We attribute this to possible constructive interference of the optical noise between stack three and fiber instance four. Figure 8b shows that the slope value drops substantially for stack instance four, as there is no main signal but only two random signals compared. This creates a circle shape instead of an ellipse in cross-plots, and so the $\lambda_{2}$ value can not describe the scenario anymore. Therefore, everything depends on the slope for this specific case. All slope values in the highpass filtered signal are reduced compared to the instance one, either stack or single fiber. Stacking reduced the noise in comparison to fiber one, and by it, the average amplitude of the signal is also reduced (Figure 8e).

We achieved, at stack instance five, a maximum reduction in the initial noise presented in the lowpass filter scenario (see Figure 8a, Appendix A). The calculated reduction is 20.04% ($\left(\lambda_{2}^{\left(5\right)}-\lambda_{2}^{\left(1\right)}\right)/\lambda_{2}^{\left(1\right)}$, where ${}^{\left(i\right)}$ is the stack instance). The evolution of $\lambda_{2}$ in Figure 8c is consistent with our experimental findings. $\lambda_{2}$ is stable until fiber instance five and then starts to increase, indicating more noise introduction after stack instance five.

The best stack instance (No. 5) shows smoother signals, where wave phases can be recognized easier than by only using one fiber (see Figure 9); however, the stack reduces the amplitude. By looking again at the slope and $\lambda_{2}$ values in Figure 8a, we can observe that the relation between the two variables is indirectly proportional; when noise is reduced, the ratio between the two DDSS signals increases. Higher noise in standard DDSS data increases the average amplitude, so they can get closer to values of the engineered DDSS signals (see Figure 5a). While interrogating longer distances, the back-scattered intensity decreases and mixes with the optical noise, as shown by the standard deviation against fiber instances (Figure 5b) and the $\lambda_{2}$ values of fiber one against other fiber instances (Figure 8c). The last fiber instance has the highest noise, and if it is used in the stack, the $\lambda_{2}$ value increases and the ratio (slope) between the standard and engineered DDSS signals decreases.

We derived a mathematical expression for estimating the best stack instance to use based on the total interrogated fiber length and the length of the fiber instances (see Appendix B). Theoretically, the maximum improvement due to stacking would be achieved in stack instance four instead of five. The discrepancy between the theoretical estimate and experimental findings is due to the firing repetition rate used for the laser probing the fiber, also known as the laser rate. The entire length from the spliced fiber instances was interrogated with the same laser rate. However, in the theoretical scenario, the maximum laser rate changes with respect to the total fiber length to optimize the survey [21], and the length changes with the fiber instances in consideration. Further, the laser rate was chosen in accordance with the time decimation applied to it to always deliver 1 kHz of output data. Therefore, we conclude that a stack of four fibers would have been the ideal case as long as the laser pulse rate is increased according to the length nearly equivalent to four fiber instances.

The PCA method was able to establish a relationship between the two fibers and describe the noise level. The example used to study the relationship is non-saturated in either of them, as they correspond to a weak tremor. However, in saturated signals, the relationship would pass from a linear to a non-linear regime, and so PCA becomes unreliable for establishing a relationship. In such a scenario, other methods should be explored to avoid non-linear regimes due to saturations.

### 5.3. Saturation

Signal saturation in DDSS is an effect due to the limitation of unwrapping for phase-shifts in the range −π<ΔΦ<π, and so this leads to a phase jump in the raw data [21]. Our phase detection algorithm recognizes phase jumps and perform a reconstruction by shifting the values in the saturated signal window near the value before the saturation occurs.

When assessing correction for saturated signals, the engineered DDSS signal was properly reconstructed, while the standard signals from stack instance five and fiber instance one were not (see Figure 10b,c). Despite the fact that there is a noise reduction due to stacking, it is not sufficient enough for the algorithm to discriminate clear phase jumps (purple curve in Figure 10d).

A phase jump described with few time samples (ideally two) located in the upper and lower limits of the dynamic range produces a clear peak in the characteristic function (CF), as in the case of the engineered DDSS signal (Figure 10e). We noticed that the time samples in phase jumps are higher in standard DDSS data (fiber instance one and stack instance four) than in the engineered ones. Phase jumps involving more than two time samples can increase the number of points involved in a CF peak. Since the algorithm relies on an accumulative summation of CF peaks over a certain threshold, the phase-jump counts could be higher. This results in a false correction with the signal moved out from the main trend, as shown in Figure 10a,b. In the end, high time samples and optical noise in phase jumps diminish the amplitudes of the CF-related peaks, and so the CF peaks look similar to the ones produced by noise. Further, high noise can make the phase jump occur in a smaller interval than the dynamic range, which also results in lower peaks in the CF (Figure 10d).

Due to the limitations of saturation correction, there are some variables to consider in order to avoid this phenomenon or facilitate the correction in standard DDSS data in the future. The initial laser rate should be increased as it can lower the noise floor to a factor of $1/\sqrt{N}$ [21], where *N* is the increment ratio of the laser rate. Additionally, to avoid the saturation effects, the sampling frequency should also be increased. The dynamic range is dependent on the sampling frequency and the gauge length; higher sampling rate and lower gauge length would increase the dynamic range, and so avoid signal saturation. However, this would reduce the strain-rate sensitivity [21,27]. Strong perturbations behave as high-frequency signals, and so it has a proportional effect on the maximum strain rate detectable. Therefore, measuring them without distortion, would require an increase in the sampling rate [22].

The conversion factor from counts to strain rate provided with the unit obeys the following equation:(3)$${\dot{\epsilon }}_{xx}=counts\times \left[\frac{116\times 10^{-9}\left(m/\mathrm{rad}\right)\times f_{s}\left(\mathrm{Hz}\right)}{L_{G}\left(m\right)\times No.}\right]$$
where $116\times 10^{-9}$ is the elongation per radian, $f_{s}$ is the sampling frequency, $L_{G}$ is the gauge length, and No.=8192 is factor associated to digitization.

Based on the strain-rate conversion factor equation, a sampling frequency of no less than 3054 Hz would have avoided signal saturation for the engineered fiber without changing the gauge length. This calculation takes into consideration a limit of 90% of the dynamic range and the maximum integer value of an int16 number type as a precaution near the real limit.

## 6. Conclusions

We demonstrated that optical noise could be reduced significantly by stacking the first two fiber instances of a standard multi-fiber cable. In our setup, a stack of five fibers achieves the maximum noise reduction of 20.04% within the main frequency content of the signal. As a result, the wavefield is smoothed, and wave phases can be easily followed. A stack of four fibers would achieve the best improvement for a sampling frequency of 1 kHz if the total interrogated fiber length has been equivalent to four spliced fiber instances of 1.5 km each. Stacking improvement depends on the cable length, which has an effect on the maximum laser rate for use. Therefore, for future multi-fiber surveys, we recommend taking these variables into consideration to establish which instance (total length) the fiber must be interrogated to achieve the best stack improvement.

Here we characterized for the first time two types of distortions in DDSS signal: quasi-phase jumps and phase jumps. The phase jumps are DDSS saturation that occur due to exceeding the dynamic range of measurement. These are characterized by phase jumps that can be corrected if SNR is large enough. We developed an algorithm that is able to correct signal saturation and restore the true values from distorted signals with some limitations, such as noise level. It is recommended to increase the laser rate as much as the total fiber length can handle to reduce the noise floor. However, this might bring limitations toward maximum time decimation that determines the output sampling frequency and, therefore, the output volume of the data.

If the optical noise is low enough, we suggest including a saturation correction scheme as part of fundamental pre-processing when working with strong motion events for DDSS measurements. Otherwise, depending on the conversion factor from counts to strain rate handled by the unit, it is advisable to estimate the range of strain or strain rate to select which final sampling frequency after time decimation and gauge length is convenient for the survey. The correct parameters would set the ideal dynamic range and sensitivity to capture the desired events and so signal saturation can be avoided for the desired events to record.

The PCA method is suitable for evaluating the relation between two signals only when the relation is mostly linear. The method also has problems when it involves saturated regimes as it becomes non-linear. Both types of fiber saturation present a non-linear regime within the record, and so machine learning approaches may help to overcome this and could also correct the quasi-phase jumps at the same time.

Despite engineered fibers having a lower noise level due to scattered points uniformly distributed along the fiber, the stacking of DDSS data on standard fibers can be a cheap option for measuring the strain rate in volcano-tectonic environments; with a noise reduction that facilitates the continuity in wave phases. Stacking DDSS data routines can be a step further and motivates the exploitation of the existing coverage of standard multi-fiber telecom cables in cities and out of urban areas.

## Figures and Tables

**Figure 1 sensors-23-03735-f001:**
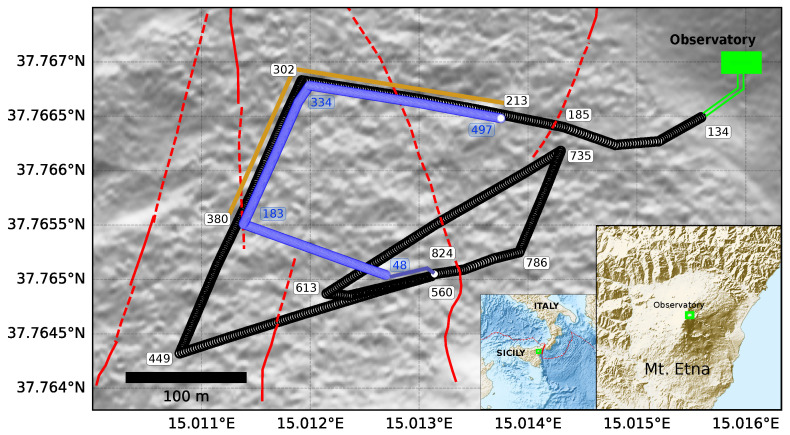
Configuration of the standard and engineered fibers at the NE of the main craters of Mt. Etna, Sicily. The two used interrogators (iDAS and Carina) are located inside the Pizzi Deneri (PDN) Observatory in green. The standard fiber with its channels is labeled in black-bordered circles, while the engineered one is labeled in blue. The orange line delimits the section of interest where the standard and engineered fibers share a similar geometrical arrangement, referred to as the common path. Known fault locations are displayed with solid red lines and inferred ones in dashed red lines.

**Figure 2 sensors-23-03735-f002:**
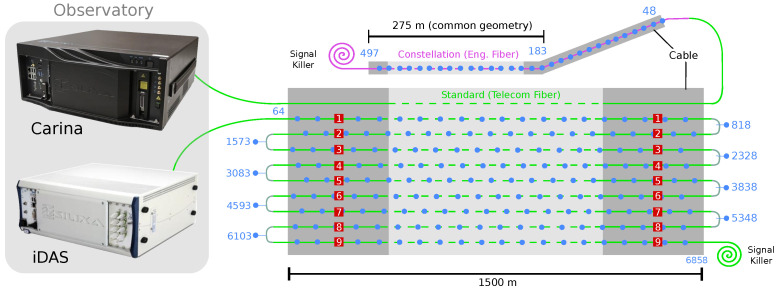
Schematic representation of the configuration used in the experiment. Grey boxes represent the cable. The green and purple lines represent the standard and engineered fibers, respectively. Individual channels are indicated by blue dots and labels. Red boxes with numbers indicate the fiber instance within the standard cable once the individual standard fibers were spliced in a back-and-forward way. Signal killers are optical attenuators made of tight coils in order to avoid high reflections from the end of the fibers. The two interrogators (iDAS and Carina) are located in the PDN Observatory.

**Figure 3 sensors-23-03735-f003:**
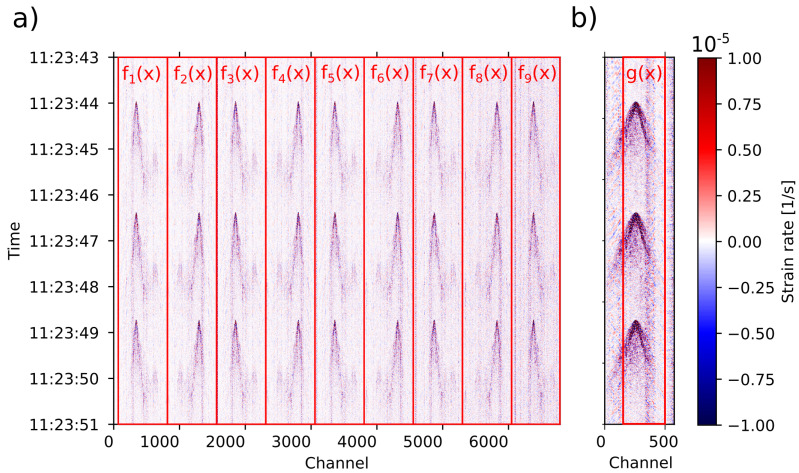
Raw data for (**a**) standard and (**b**) engineered fiber showing signals of tap tests. The data were divided into N=9 windows that represent the fiber instances of Figure 2, while g(x) represents the defined section of the engineered fiber. The signals shown were used only for the purpose of finding the channel correspondence between the fiber instances of the standard fiber and between standard and engineered fibers.

**Figure 4 sensors-23-03735-f004:**
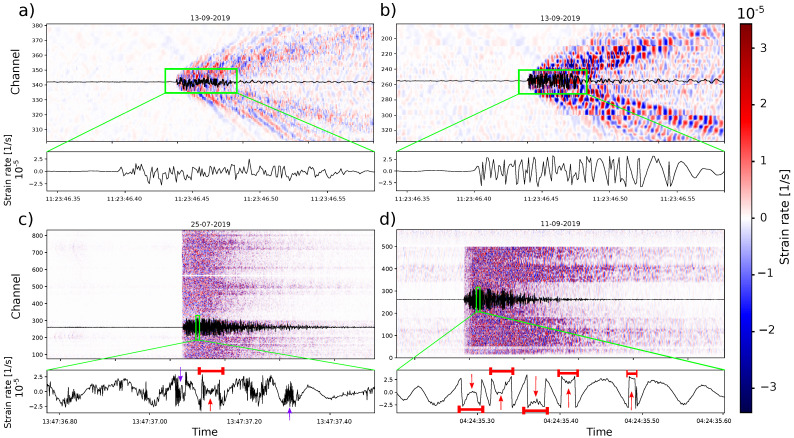
Example of DDSS signals for specific channels in: (**a**) standard fiber for a tap test signal, (**b**) engineered fiber for a tap test signal, (**c**) standard fiber for a local earthquake signal, and (**d**) engineered fiber for a local earthquake signal. In each subfigure, the top part shows the DDSS data across channels, with the waveform of one selected channel, and on the bottom, a zoomed version of the same signal, indicated by the green square. For standard and engineered DDSS data, the tap test signal is the same (10 Hz) and is shown in one part of the common section. However, the local earthquake is not (6 and 10 Hz for (**c**,**d**), respectively). Subfigures (**c**,**d**) show indications of recognized major phase jumps (red arrows) and minor phase jumps (purple).

**Figure 5 sensors-23-03735-f005:**
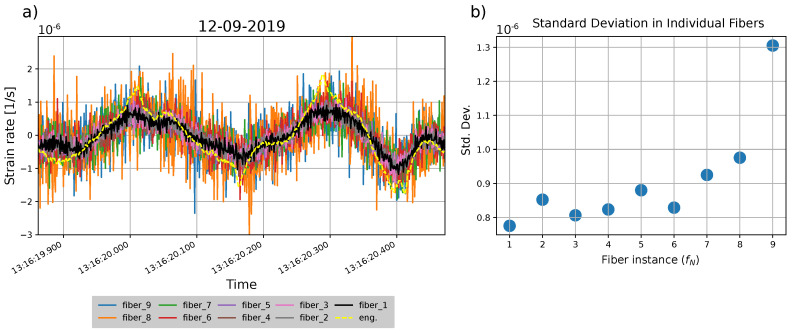
Fiber instance comparisons. (**a**) Raw signal from a volcanic tremor from Mt. Etna, recorded at equivalent channels between the standard fiber instances and the engineered fiber. For raw data, each fiber is labeled with its specific color below. The equivalent channels are 336 and 261 for standard fiber one and engineered fiber, respectively. Other equivalences between the other fibers can be found in the Appendix A. (**b**) Standard deviation values (blue dots) at each standard fiber instance for a defined time window of the volcanic tremor. All the length of the fiber was used instead of the shared section with the engineered fiber.

**Figure 6 sensors-23-03735-f006:**
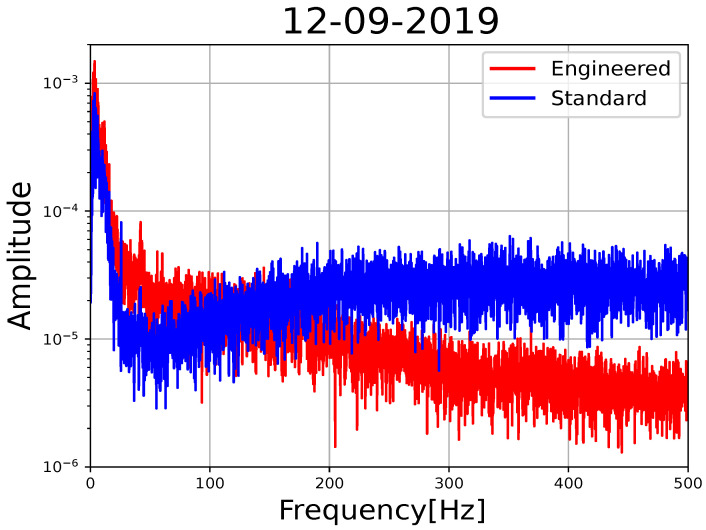
Smoothed spectrum of standard fiber (fiber instance one) and engineered fiber in blue and red curves, respectively. The smoothing is performed by taking only the prominent peaks of each original spectral curve. Original curves are shown in the Appendix A.

**Figure 7 sensors-23-03735-f007:**
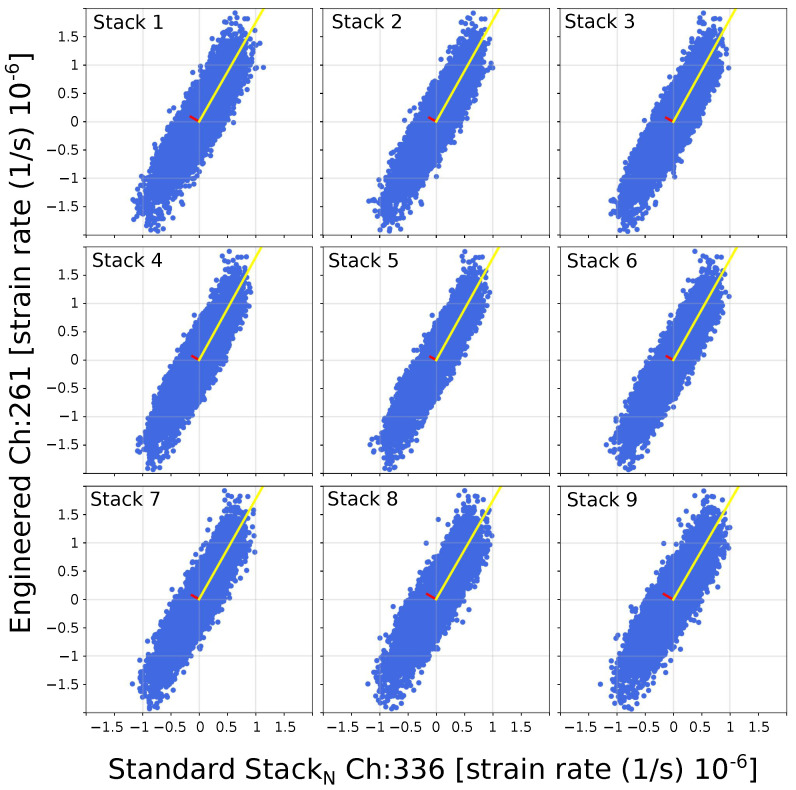
Strain-rate cross-plots between the stack instances of the standard fibers and engineered fiber amplitudes using the equivalence of channels 336 (standard) and 261 (engineered). The yellow and red lines indicate the direction of the eigenvectors $\vec{e_{1}}$ and $\vec{e_{2}}$ for the specific stack instance vs. engineered amplitude comparison, respectively.

**Figure 8 sensors-23-03735-f008:**
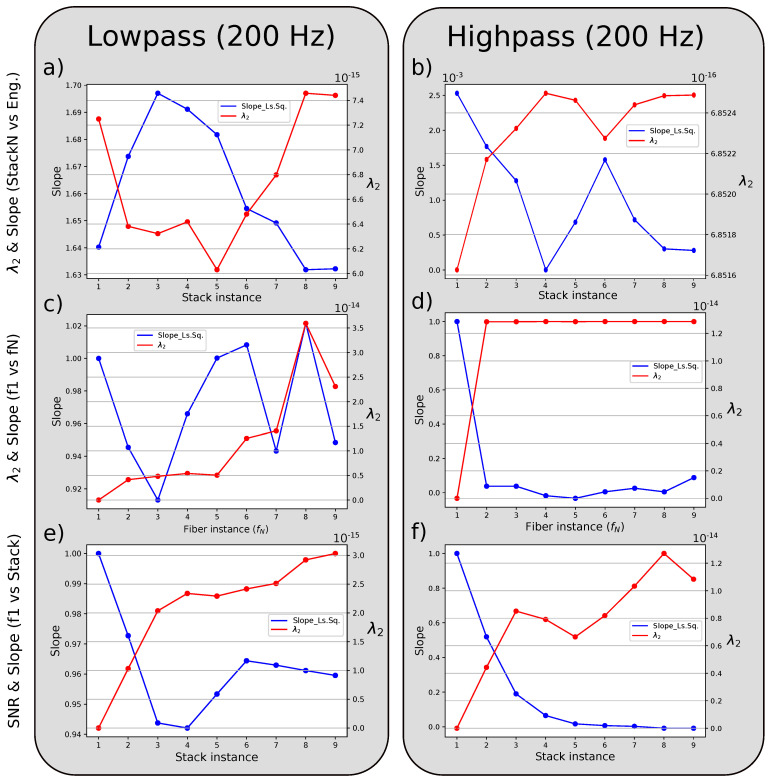
Calculated slope (blue curve) and $\lambda_{2}$ (red curve) from the PCA method for the tremor signal with an applied lowpass and highpass filters (200 Hz) at each of the following scenarios: standard stack instances vs. engineered fiber (**a**,**b**), standard fiber instance one vs. other standard instances (**c**,**d**), and standard fiber instance one vs. standard stack instances (**e**,**f**).

**Figure 9 sensors-23-03735-f009:**
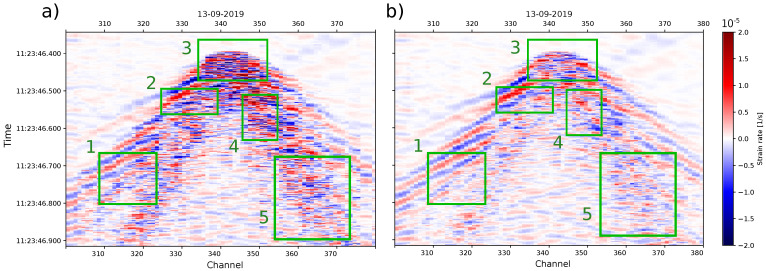
DDSS data example of a tap test (same as in Figure 4) for (**a**) fiber instance one and (**b**) stack instance five. Noticeable differences are indicated within the green boxes where the signal has been smoothed and due to random noise reduction.

**Figure 10 sensors-23-03735-f010:**
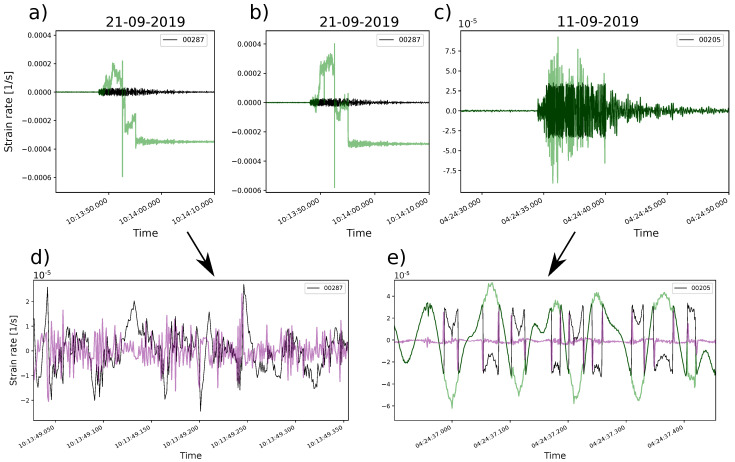
Signal saturation correction applied to single-channel signals for: (**a**) stack instance five, (**b**) fiber instance one (stack instance one), and (**c**) engineered fiber. Sub-figures (**d**,**e**) show zoomed sections of (**a**,**b**), respectively. Black curves are the raw signal, green curves are the corrected signals, and purple curves are the characteristic function (CF) previously defined in Section 3.

## Data Availability

The data are available by request to the authors and will be put online at the GEOFON data repository after an embargo period of 3 years (2026).

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
