# Peer review of "On the Comparison of Records from Standard and Engineered Fiber Optic Cables at Etna Volcano (Italy)"

_sensors, 2023, doi:10.3390/s23073735_

Round 1
Reviewer 1 Report
Please see attached

Author Response
Response to Reviewer 1 Comments
The authors investigate methodology of Distributed Acoustic Sensing and consider a new way of improving Signal-to-Noise Ratio (SNR) via utilizing several fibers incorporated in the same optical cable.
For reference, they use a and “engineered” Constellation fiber that exhibits ~20 dB stronger Rayleigh scattering. Ideally, each “regular” fiber from the cable should produce the same response to acoustic or seismic wave, so that if N fibers are used, the SNR should become ~√? times larger. However, since the fibers are a few km long, it is difficult to correlate the points that correspond to the same acoustic wave phase, which makes the task of improving the SNR not that straightforward. The authors applied a series of mathematical techniques for correlating the adjacent fibers and analyzing pros and cons of their approach. The performed study is important from a practical point of view. The technical level is high, the English is good, and the work deserves a publication in Sensors.
Thank you for your positive feedback.
I recommend making just minimal edits:
• Page 5, Eq. 1. Maybe it would be easier to comprehend this equation, if the SCC is denoted as
SCC(1, 2).
We have introduced this change in the equation. In addition, we introduced a new term, expanding the equation to 3 terms with two equal signs. We believe that this helps to follow the meaning of stacking cross-correlation functions, and between which spatial functions are involved in cross-correlation function.
- Page 8, line 226. The word “fiber” is missing between “engineered” and “diminishes”.
Done.
- Pages 8 and 9. The Principle Component Analysis may not be familiar for most of the readers. I suggest adding a 2-3 introduction sentences to this method and also explaining the terms “l2” and “slope”.
The guidelines of the journal indicate that Methods should come before Results. Therefore, in Methods (Section 3.3 “Standard and engineered channel equivalences”), we introduced the Principal Component Analysis, and what we could interpret from the resulted values.
Nevertheless, we introduced 2 changes in Page 8 and 9, so the reader be reminded of the meaning of the terms.
- At Page 8, Line 231, we added a parenthesis which points to section 3.3 for more description of the method.
- At Page 9, Paragraph 1, we reminded of the meaning of the terms lambda_2 and slope by modifying the paragraph with more information.
- Page 14, line 376. The authors should explain what “laser rate” means.
Agreed. We introduced this change on the first time we mentioned laser rate, on Page 13, Line 337-338.
Reviewer 2 Report
The authors present an interesting work from a practical point of view. They analysed how stacking can improve the signal quality. The authors found out that the stack of DDSS records from 5 standard fiber instances, 1.5 km long each, can reduce optical noise up to 20%. They also presented an algorithm to correct artifacts in the time series that stem from dynamic range saturation. They claim that the algorhithm although stacking is able to reduce optical noise, it is not sufficient for restoring the strain-rate amplitude from saturated signals in standard fiber DDSS. All these things are significant and important in this field of study. However, I have a few remarks which must be corrected before this manuscript is published.
1. While the specification of a standard single-mode optical fiber for telecommunications is freely available and well known, the authors do not provide the basic parameters of the special fiber also involved in the experiment. Any scientific research must be repeatable, and without knowledge of these parameters it is impossible. I recommend the authors to add this information to the article.
2. I'm a little confused on lines 88-89. The authors use two different types of optical fibers and two different interrogators. Carina is connected to a special optical fiber, iDAS is connected to a telecommunication cable. However, in the mentioned lines it is written that an engineered fiber is connected to one of the standard fibers and a link to Figure 2 is provided. In this case, in Figure 2, fibers of different types are not connected to each other. First, it introduces confusion. Secondly, does this mean that, when comparing two measurements, the authors assume that the characteristics of the interrogators are the same? Why weren't these two fibers connected in line and tested in turn with two different systems? I recommend that the information in this section be given in a more detailed and understandable way.
3. I recommend making the same clarity on lines 100-102.
4. It is necessary to explain the physical meaning of figure 5b. The x-axis, as described, represents the fiber number (in the cable). In this case, the points on the graph are connected, which indicates some dependence. I would like to ask you to describe this figure in more detail.
5. Figure 6 appears before its first mention in the text.
6. In Figure 6, the transparency of the spectrum corresponding to the standard fiber gives the false impression that three spectra are shown instead of two.
7. Figure 7 appears before its first mention in the text.
8. Another question concerns the need to conduct such a comparison on a natural object, which has a limited range of vibration frequencies. The authors need to explain why, instead of a laboratory experiment, they prefer an oundoor test. In the laboratory, the one can set a wide range of frequencies and apply them to the desired fiber/cable location. Perhaps it would be more informative in terms of demonstrating the method. I would like the authors to say a few words about this at the beginning of the work.
Author Response
The authors present an interesting work from a practical point of view. They analysed how stacking can improve the signal quality. The authors found out that the stack of DDSS records from 5 standard fiber instances, 1.5 km long each, can reduce optical noise up to 20%. They also presented an algorithm to correct artifacts in the time series that stem from dynamic range saturation. They claim that the algorhithm although stacking is able to reduce optical noise, it is not sufficient for restoring the strain-rate amplitude from saturated signals in standard fiber DDSS. All these things are significant and important in this field of study. However, I have a few remarks which must be corrected before this manuscript is published.
Thanks for your positive feedback. We address each of your comments below.
While the specification of a standard single-mode optical fiber for telecommunications is freely available and well known, the authors do not provide the basic parameters of the special fiber also involved in the experiment. Any scientific research must be repeatable, and without knowledge of these parameters it is impossible. I recommend the authors to add this information to the article.
We introduced a new paragraph in section 2 “Material and Setup”, below Figure 2, that describes more the design of the used engineered fiber, alongside with citations of both American and European patents that contains the description.
- I'm a little confused on lines 88-89. The authors use two different types of optical fibers and two different interrogators. Carina is connected to a special optical fiber, iDAS is connected to a telecommunication cable. However, in the mentioned lines it is written that an engineered fiber is connected to one of the standard fibers and a link to Figure 2 is provided. In this case, in Figure 2, fibers of different types are not connected to each other. First, it introduces confusion. Secondly, does this mean that, when comparing two measurements, the authors assume that the characteristics of the interrogators are the same? Why weren't these two fibers connected in line and tested in turn with two different systems? I recommend that the information in this section be given in a more detailed and understandable way.
Indeed, Figure 2 shows how the connection were made. The iDAS unit sampled the standard fiber (spliced fibers). The Engineered fiber was spliced to one of the standard fibers, as shown in the image, and this was connected to a Carina unit. We corrected the first sentence, so instead of “connected“, we used “recorded“ to make clear that each fiber was interrogated with a different unit. We also added at Line 102-103 a reminder of how despite the connection of the standard fiber to the Carina unit, only the section with the engineered fiber was recorded.
- I recommend making the same clarity on lines 100-102.
Addressed in the point 2.
- It is necessary to explain the physical meaning of figure 5b. The x-axis, as described, represents the fiber number (in the cable). In this case, the points on the graph are connected, which indicates some dependence. I would like to ask you to describe this figure in more detail.
Since the fiber instances are spliced, as we increase in the number, we also increase in distance over the total fiber length. Each fiber is 1.5 Km long (cable length), and so, fiber instances can be interpreted as distance D = N * 1.5 Km, where N is the fiber instance number. However, as each computation of the signal RMS is performed individually for each fiber instance, there is indeed no dependency. Therefore, we took out the line connecting the dots, and enhanced the dots. We also described the Figure in more detail with a new caption.
- Figure 6 appears before its first mention in the text.
This happens in other figures too. We decided to do this so the document looks esthetically good. LaTex arranges the figures and paragraphs differently as they appear in the code to fit them in the pages. Therefore, paragraphs might be set at different positions compared to the figures that they refer to. Nevertheless, we decide to address the reviewer’s comment and take care of this by applying this suggestion to all of the figures. The editorial office is solely responsible on this issue.
- In Figure 6, the transparency of the spectrum corresponding to the standard fiber gives the false impression that three spectra are shown instead of two.
Both spectral curves are displayed in the same graph for comparison. The spectral curves cannot be displayed without transparency, otherwise both of their forms cannot be fully seen. Nevertheless, we found a way to present the curve without the transparency. The new figure shows smoothed curves by generating new spectral curves from the prominent peaks of each one. The caption was changed so it refers to the raw spectral curves, now located in Supplementary Material.
- Figure 7 appears before its first mention in the text.
Addressed in point 5.
- Another question concerns the need to conduct such a comparison on a natural object, which has a limited range of vibration frequencies. The authors need to explain why, instead of a laboratory experiment, they prefer an oundoor test. In the laboratory, the one can set a wide range of frequencies and apply them to the desired fiber/cable location. Perhaps it would be more informative in terms of demonstrating the method. I would like the authors to say a few words about this at the beginning of the work.
It is true that experiments carried in a laboratory would have been more controlled, and we could apply a wide range of frequencies on fiber locations. However, we selected Etna for 2 reasons.
- Mt. Etna has been widely studied, and there is a consensus of how the natural seismic sources provide a wide frequency range. Also, the spectral curves show that the tremor signal also had high-frequency signals embedded within. This was also confirmed by inspection of the raw data for 2 equivalent channels between the 2 types of fibers. Therefore, the range of frequencies is not limited.
- We wanted to carry this study outdoors due to the complex natural seismic environment and the increase interest of applied DDSS (a.k.a. DAS) on natural hazards, specially volcanology and tectonics. We believe that readers could be more interested to see the effects in an applicable scenario such as volcano-seismology, and show that this could work in complex seismo-acoustic environments.
Reviewer 3 Report
Authors provide a comprehensive comparison between the standard fiber and the engineered fiber optical cables. The topic is very interesting and the manuscript is well organized. I only have two small comments.
1. is the engineered fiber used in this work helical or not? please clarify it.
2. please explain why the authors choose a 2m for standard fiber and 1m for the engineered fiber. is it related to the channel matrix size?
Author Response
Authors provide a comprehensive comparison between the standard fiber and the engineered fiber optical cables. The topic is very interesting and the manuscript is well organized. I only have two small comments.
Thank you for your interest and your positive feedback.
- is the engineered fiber used in this work helical or not? please clarify it.
Neither the standard and engineered fibers are helical. We introduced a parenthesis clarifying that they are non-helical at the beginning of the section “Materials and Setup”.
- please explain why the authors choose a 2m for standard fiber and 1m for the engineered fiber. is it related to the channel matrix size?
This was a previous configuration, since this experiment is part of a larger project in Mt. Etna for volcano monitoring. The standard fiber was set to sample with a spatial interval of 2 meters. We could only interrogate the engineered fiber for a couple of days, and so our initial goal was to exploit the performance of the engineered fiber by setting up a finer spatial interval than the standard fiber. Nevertheless, we would like to remind that the difference in spatial interval does not have any effect on the time-series data. Additionally, it does not have any effect on the spatial cross-correlation results, as shown by the analysis performed, either introducing artificial channels in standard fiber data, or removing intermediate channels on engineered fiber data.